# Unsupervised Diffusion and Volume Maximization-Based Clustering of Hyperspectral Images

**Sam L. Polk** [1], **Kangning Cui** [2], **Aland H. Y. Chan** [3,4], **David A. Coomes** [3,4], **Robert J. Plemmons** [5] **and James M. Murphy** [1,*]

1. Department of Mathematics, Tufts University, 177 College Ave., Medford, MA 02155, USA
2. Department of Mathematics, City University of Hong Kong, 83 Tat Chee Ave., Kowloon, Hong Kong
3. Conservation Research Institute, University of Cambridge, Downing Street, Cambridge CB2 3EA, UK
4. Department of Plant Sciences, University of Cambridge, Downing Street, Cambridge CB2 3EA, UK
5. Departments of Mathematics and Computer Science, Wake Forest University, 1834 Wake Forest Rd., Winston-Salem, NC 27109, USA
* Correspondence: jm.murphy@tufts.edu

**Abstract:** Hyperspectral images taken from aircraft or satellites contain information from hundreds of spectral bands, within which lie latent lower-dimensional structures that can be exploited for classifying vegetation and other materials. A disadvantage of working with hyperspectral images is that, due to an inherent trade-off between spectral and spatial resolution, they have a relatively coarse spatial scale, meaning that single pixels may correspond to spatial regions containing multiple materials. This article introduces the Diffusion and Volume maximization-based Image Clustering (D-VIC) algorithm for unsupervised material clustering to address this problem. By directly incorporating pixel purity into its labeling procedure, D-VIC gives greater weight to pixels corresponding to a spatial region containing just a single material. D-VIC is shown to outperform comparable state-of-the-art methods in extensive experiments on a range of hyperspectral images, including land-use maps and highly mixed forest health surveys (in the context of ash dieback disease), implying that it is well-equipped for unsupervised material clustering of spectrally-mixed hyperspectral datasets.

**Keywords:** hyperspectral imaging; clustering; diffusion geometry; spectral unmixing; forest health; ash dieback





## 1. Introduction

Hyperspectral images (HSIs) are images of a scene or object that store spectral reflectance at a hundred or more spectral bands per pixel [1–3]. HSI remote sensing data, which is generated continuously by airborne and space-borne sensors, has been used successfully for signal processing problems in fields including forensic medicine (e.g., age estimation of forensic traces [4]), conservation (e.g., species mapping in wetlands [5,6]), and ecology (e.g., estimating water content in vegetation canopies [7]). The high-dimensional characterization of a scene provided in remote sensing HSI data has motivated its use in material classification problems [8], wherein machine learning is used to separate pixels based on the constituent materials (including vegetation types, trees species, and plant health) within spatial regions [2,3,9].

Though hyperspectral imagery has become an essential tool across many scientific domains, material classification using HSI data faces at least two key challenges. First, because of an inherent trade-off between spectral and spatial resolution, HSIs are generated at a coarse spatial resolution [10–15]. One would prefer an HSI with both a high spatial resolution (so that individual pixels correspond to spatial regions containing just one material) and a high spectral resolution (to enable capacity for material classification) [11]. However, an increase in the spatial resolution of an HSI often comes at the cost of reducing the effective detection energy entering the recording spectrometer across each spectral

band [10]. While this effect may at least partially be mitigated by increasing the aperture of the optical system underlying the spectrometer used to generate HSI data [16], high-aperture instruments generally also have high volume and weight [10]. As such, HSI data is typically generated at a coarse spatial resolution (roughly 1 m from drone, 3–10 m from aircraft, 30 m from space). Thus, though some high-purity pixels in an HSI may correspond to spatial regions containing predominantly just one material, other pixels are mixed, corresponding to spatial regions with many distinct materials [12,14]. A second challenge is that the generation of expert labels—often used for training supervised machine learning models—is generally impractical due to the large quantities of HSI data continuously produced by remote sensors [1]. To efficiently analyze unlabeled HSIs, one may use HSI clustering algorithms, which partition HSI pixels into groups of points sharing key commonalities [17]. These algorithms are unsupervised; i.e., ground truth labels are not used to provide a partition of an HSI [17]. Though clustering has become an important tool in the field of hyperspectral imagery [18–31], HSI clustering algorithms that do not directly account for the fact that HSI pixels are often spectrally mixed may fail to extract meaningful latent cluster structure [32,33].

This article introduces the Diffusion and Volume maximization-based Image Clustering (D-VIC) algorithm for unsupervised material classification (i.e., material clustering) of HSIs. D-VIC is the first algorithm to simultaneously exploit the high-dimensional geometry [19,34] and abundance structure [12,32] observed in HSIs for the clustering problem. In its first stage, D-VIC locates cluster modes: high-purity, high-empirical density pixels that are far in diffusion distance (a data-dependent distance metric [35]) from other high-purity, high-density pixels. These pixels serve as exemplars for all underlying material structures in the HSI. In its mode selection, D-VIC downweights high-density pixels that correspond to commonly co-occurring groups of materials. As such, D-VIC's exploitation of spectrally mixed structure in HSI data [10–13] enables the selection of modes that better represent the material structure in the scene. After detecting cluster modes, D-VIC propagates modal labels to non-modal pixels in order of decreasing density and pixel purity. Since pixel purity is also incorporated into D-VIC's non-modal labeling, D-VIC accounts for material abundance structure in the HSI during its entire labeling procedure. D-VIC is compared against classical and related state-of-the-art HSI clustering algorithms on three benchmark real HSI datasets and applied to the problem of unsupervised detection of a forest pathogen—ash dieback disease (*Hymenoscyphus fraxineus*) [36–39]—using real remote sensing HSI data. On each dataset, D-VIC produces competitive unsupervised labelings and, moreover, enjoys robustness to hyperparameter selection. Computationally, D-VIC scales quasilinearly in the size of the HSI, and its empirical runtime is competitive, suggesting it is well-suited to cluster large HSIs.

The rest of this article is structured as follows. Section 2 provides background on HSI clustering, diffusion geometry, and spectral unmixing. Section 3 motivates incorporating spectral unmixing into a nonlinear graph-based clustering framework and introduces D-VIC. Section 4 demonstrates the efficacy of D-VIC through substantial experiments on three real HSI datasets. Additionally, it is shown in Section 4 that D-VIC may be used for an unsupervised ash dieback disease detection problem using remotely sensed HSI data collected over a forest in Great Britain [40]. We conclude and offer directions for future work in Section 5. Finally, in Appendix A, we detail hyperparameter optimization.

## 2. Background

### 2.1. Background on Unsupervised HSI Clustering

HSI clustering algorithms partition an HSI, denoted $X = \{x_i\}_{i=1}^{n} \subset \mathbb{R}^D$ (interpreted as a point cloud of HSI pixels' spectral signatures, with $n$ pixels and $D$ spectral bands) into $K$ clusters of pixels. The partition, which we call a clustering of $X$, may be encoded in a labeling vector $\mathcal{C} \in \{1, 2, \ldots, K\}^n$ such that $\mathcal{C}(x_i) = \mathcal{C}_i \in \{1, 2, \ldots, K\}$ is the label assigned to the pixel $x_i$. Ideally, pixels from any one cluster are in some sense "related," and pixels from any two clusters are "unrelated" [17,28,41,42]. Clustering algorithms

are unsupervised, meaning that data points are labeled without the aid of any expert annotations or ground truth labels. This has motivated the development of algorithms explicitly built for material clustering using HSIs [18–31,43].

Though classical clustering algorithms such as *K*-Means and the Gaussian Mixture Model (GMM) [17] remain widely used in practice, these algorithms tend to perform poorly on HSIs for a number of reasons [18,19]. First, algorithms that rely on Euclidean distances are prone to the "curse of dimensionality" on datasets like HSIs with a high ambient dimension (i.e., the number of spectral bands is large) [44]. Second, HSIs are often spectrally mixed [10–13], and overlap may exist between clusters in Euclidean space [18]. A final complication is that classical algorithms generally assume that latent clusters in a dataset are approximately ellipsoidal groups of points that are well-separated in Euclidean space [17], but clusters in HSIs often exhibit nonlinear structure [34]. A simple toy dataset, visualized in Figure 1 [41], serves as an example of a dataset with a nonlinear structure. This dataset lacks a linear decision boundary between its $K = 2$ latent clusters, and classical algorithms (*K*-Means and GMM [17]) could not learn its latent nonlinear cluster structure. HSIs often contain clusters that can only be separated using a nonlinear decision boundary [18]; thus, algorithms that rely solely on Euclidean distances are expected to perform poorly at material clustering on HSIs.

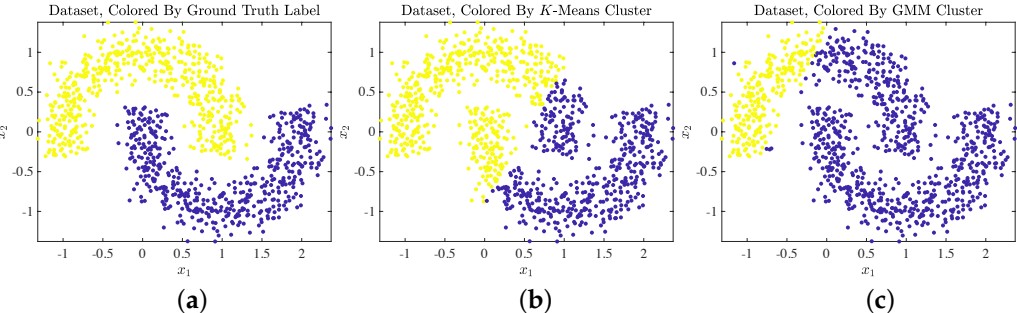

**Figure 1.** Example of a toy dataset ($n = 1000$) with nonlinear cluster structure: two interleaving half-circles. The idealized clustering, visualized in (**a**), separates each half-circle. Due to the lack of a linear decision boundary, however, both *K*-Means (**b**) and GMM (**c**) were unable to extract latent cluster structure from this simple nonlinear dataset. Both algorithms were run with 100 replicates.

The limitations outlined above have motivated the application and development of nonlinear graph-based algorithms for HSI clustering [18–24,31,41,43,45–47]. Graph-based algorithms rely on data-generated graphs; pixels are represented as nodes in the graph, and edges encode pairwise similarity between them. Highly connected regions in the graph may then be summarized using a nonlinear coordinate transformation [35,48–50], as is described in more detail in Section 2.2. Thus, a partition may be obtained by implementing a classical clustering algorithm on the dimension-reduced dataset. Due to their reliance on a graph representation of an HSI, these algorithms tend to be robust to small perturbations in the data and noise. Moreover, theoretical guarantees exist for the successful recovery of latent cluster structure, even if boundaries between latent clusters are nonlinear [42,51,52]. Despite their exhibited successes, algorithms that rely solely on graph structure tend to perform poorly on datasets containing multimodal cluster structure [42,52,53]; i.e., if a single cluster has multiple regions of high and low density. Importantly, this includes spectrally mixed HSIs, the classes of which often contain multiple co-occurring materials of varying abundances [10–12].

Deep neural networks and graph convolutional networks have recently become popular for material classification and clustering in HSIs because of their capacity to predict complex data sources [25–27,29,30,47,54,55]. While these algorithms tend to be highly accurate at material classification using real HSI data, many state-of-the-art deep models for HSI segmentation still rely in some part on training labels, whether via pre-training some or all of the network [54,55] or explicitly relying upon a small number of ground

truth labels [25,26,47,56], and/or pseudo-labels [25,29]. Moreover, even fully unsupervised "deep clustering" algorithms [27,30] rely on deep neural networks, which have been shown to be prone to error from perturbations and noise [57,58] and whose success in unsupervised clustering is often due to data pre-processing steps rather than the unsupervised network learning meaningful features [59].

### 2.2. Background on Spectral Graph Theory

As overviewed in Section 2.1, graph-based clustering algorithms learn latent, possibly nonlinear cluster structure from HSIs by treating pixels as nodes in an undirected, weighted graph, where connections between pixels are encoded in a weight matrix $\mathbf{W} \in \mathbb{R}^{n \times n}$ [41,42,52,60]. In large datasets like HSIs, edges can be restricted to the first $N \ll n \ \ell^2$-nearest neighbors (i.e., Euclidean distance nearest neighbors) and given unit weight. In other words, $\mathbf{W}_{ij} = 1$ if $x_i$ is one of the $N$ nearest neighbors of $x_j$ or vice versa, and $\mathbf{W}_{ij} = 0$ otherwise. Let $\mathbf{P} = \mathbf{D}^{-1}\mathbf{W}$, where $\mathbf{D} \in \mathbb{R}^{n \times n}$ is the diagonal degree matrix defined by $\mathbf{D}_{ii} = \sum_{j=1}^{n} \mathbf{W}_{ij}$. The matrix $\mathbf{P} \in \mathbb{R}^{n \times n}$ may be interpreted as the transition matrix for a Markov diffusion process on $X$ and has a unique stationary distribution $\pi \in \mathbb{R}^{1 \times n}$ satisfying $\pi\mathbf{P} = \pi$ [35,42]. Define $\{(\lambda_i, \psi_i)\}_{i=1}^{n}$ to be the (right) eigenvalue-eigenvector pairs of $\mathbf{P}$, sorted in non-increasing order so that $1 = \lambda_1 > |\lambda_2| > \cdots > |\lambda_n| \geq 0$. The first $K$ eigenvectors of $\mathbf{P}$ often concentrate on the $K$ most coherent subgraphs in the graph underlying $\mathbf{P}$, making these vectors useful for clustering [41].

### Background on Diffusion Geometry

Diffusion distances are a family of data-dependent distance metrics which enable comparisons between points in the context of the Markov diffusion process encoded in $\mathbf{P}$ [35]. Diffusion distances have been successfully used in a number of applications (e.g., in gene expression profiling [61,62], data visualization [63,64], and molecular dynamics analysis [65–67]). Moreover, diffusion distances have been shown to efficiently capture low-dimensional structure in HSI data, resulting in excellent clustering performance [18,52].

Define $D_t(x_i, x_j) = \sqrt{\sum_{k=1}^{n}[(\mathbf{P}^t)_{ik} - (\mathbf{P}^t)_{jk}]^2 / \pi_k}$ to be the diffusion distance at time $t \geq 0$ between pixels $x_i, x_j \in X$ [35,68,69]. Diffusion distances are a nonlinear data-dependent distance metric that have a natural connection to the clustering problem [42,52]. To see this, note that $D_t(x_i, x_j)$ may be interpreted as the Euclidean distance between the $i^{\text{th}}$ and $j^{\text{th}}$ rows of $\mathbf{P}^t$, weighted according to $1/\pi$. If pixels from the same cluster share many high-weight paths of length $t$, but paths of length $t$ between any two pixels from different clusters are relatively low weight, then the $i^{\text{th}}$ and $j^{\text{th}}$ rows of $\mathbf{P}^t$ are expected to be nearly equal for pixels $x_i$ and $x_j$ from the same cluster and very different if these pixels come from different clusters. So, the diffusion distance between points from the same cluster is expected to be small, and the diffusion distance between points from different clusters is expected to be large [42,52]. Diffusion distances can be efficiently computed using the eigendecomposition of $\mathbf{P}$: $D_t(x_i, x_j) = \sqrt{\sum_{k=1}^{n} \lambda_k^{2t}[(\psi_i)_k - (\psi_j)_k]^2}$ [35,68,69]. For $t$ sufficiently large so that $|\lambda_k|^{2t} \approx 0$ for $k > \ell$, the sum in diffusion distances can be truncated past the $\ell^{\text{th}}$ term, yielding an accurate and efficient approximation of diffusion distances. Importantly, the relationship between diffusion distances and the eigendecomposition of $\mathbf{P}$ indicates that diffusion distances may be interpreted as Euclidean distances after nonlinear dimensionality reduction via the following dimension-reduced mapping of the ambient space into $\mathbb{R}^\ell$: $x_i \to [\lambda_1^t(\psi_1)_i \ \lambda_2^t(\psi_2)_i \ \ldots, \lambda_\ell^t(\psi_\ell)_i]$ [35,68,69].

HSIs often encode well-defined latent multiscale cluster structures that can be learned by diffusion-based HSI clustering algorithms by varying the time parameter $t$ in diffusion distances [52,60]. Indeed, smaller $t$ values generally enable the detection of fine-scale local cluster structure, while larger $t$ values enable the detection of coarse-scale global cluster structure. However, for algorithms that require $K$ as an input, $t$ must be tuned to correspond to the desired number of clusters. Thus, the choice of $t$ must be carefully considered when clustering a dataset using an algorithm that relies on diffusion distances [52,60].

### 2.3. Background on Spectral Unmixing

Real-world HSI data is often generated at a coarse spatial resolution; thus, pixels may correspond to spatial regions containing multiple materials [70–72]. To learn the latent material structure from HSIs, spectral unmixing algorithms decompose each pixel's spectrum into a linear combination of endmembers that encode the spectral signature of materials in the scene. The endmembers may be understood as "pure" material signatures. When representing a pixel as a linear combination of these endmembers, the coefficients of the linear combination indicate the relative abundance of materials within the spatial region corresponding to that pixel. Mathematically, a spectral unmixing algorithm learns $\mathbf{U} = (u_1 \, u_2 \, \dots \, u_m)^T \in \mathbb{R}^{m \times D}$ (with rows encoding the spectral signatures of endmembers) and $\mathbf{A} \in \mathbb{R}^{n \times m}$ (with rows encoding abundances) such that $x_i \approx \sum_{j=1}^m \mathbf{A}_{ij} u_j$ for each $x_i \in X$ [72]. Usually, the entries of $\mathbf{A}$ are nonnegative and normalized so that $\sum_{j=1}^m \mathbf{A}_{ij} = 1$ for each $i$; hence, abundances are data-dependent features storing estimates for the relative frequency of materials in pixels. The purity of $x_i \in X$, defined by $\eta(x_i) = \max_{1 \leq j \leq m} \mathbf{A}_{ij}$ [32], will be large if the spatial region corresponding to $x_i$ is highly homogeneous (i.e., containing predominantly just one material) and small otherwise. As such, pixel purity and spectral unmixing may be used to aid in the unsupervised clustering of HSIs [21,28,32].

Spectral unmixing has become an important tool in hyperspectral imagery, prompting its usage in a number of applications (e.g., image reconstruction [73–75], noise reduction [76–78], spatial resolution enhancement [79–81], supervised material classification [32,82,83], change detection [84–87], and anomaly detection [31,88,89]). The importance of spectral unmixing in remote sensing has motivated the development of many algorithms for this task, which we broadly summarize here; see surveys [12,90–94] for a more thorough overview. Geometric methods for spectral unmixing estimate endmembers by searching for points that form a simplex of minimal volume, subject to a constraint that at least some nearly pure pixels exist within the observed HSI pixels [12,14,19,70,95–104]. For highly mixed HSIs that lack pure pixels, statistical methods may be used [105–109]. These methods typically treat spectral unmixing as a blind source separation problem, and though they are often successful at this task, statistical algorithms are usually more computationally expensive [12]. Additionally, autoencoding methods learn latent spectral mixing structure by training neural networks that map pixel spectra to a lower-dimensional space that can be related to endmember and abundance matrices $\mathbf{U}$ and $\mathbf{A}$ [110–118]. Finally, while linear spectral unmixing is well-developed and widely used in practice, some nonlinear unmixing algorithms (including some relying on neural networks [111,115,119–122]) have been developed to account for nonlinear interactions between endmembers [111,115,119,120,122–127]. Nevertheless, many of these algorithms typically require training data or hyperparameter inputs, unlike many of the linear mixing models reviewed above [12].

Spectral unmixing is relevant to our paper as a way to determine cluster modes in an unsupervised setting. Below, we focus on two standard methods in unmixing but note that D-VIC is modular in this regard, and other unmixing algorithms could be used.

#### 2.3.1. Background on the HySime Algorithm

Hyperspectral Signal Subspace Identification by Minimum Error (HySime) is a standard algorithm for estimating the number of materials $m$ in $X$ [128]. HySime assumes that each $x_i \in X$ is of the form $x_i = y_i + \zeta_i$, where $y_i \in \mathbb{R}^D$ and $\zeta_i \in \mathbb{R}^D$ model the signal and noise associated with $x_i$, respectively. If signal vectors are linear mixtures of $m$ ground truth endmembers (i.e., $y_i = \sum_{j=1}^m \mathbf{A}_{ij} u_j$ for $1 \leq i \leq n$), then the set $\{y_i\}_{i=1}^n$ lies on a $m$-dimensional subspace of $\mathbb{R}^D$. With this motivation, HySime estimates the subspace dimension by balancing the error of projecting signal vectors $\{y_i\}_{i=1}^n$ onto their first $m$ principal components with the amount of noise captured by those vectors' orthogonal complement. Though other algorithms exist for estimating the number of materials in a scene using HSI data, many of these alternatives rely on hyperparameter inputs to estimate $m$ or have large computational complexity [129]. For example, the ubiquitous virtual dimensionality—which relies on a Neyman–Pearson detection theory-based threshold to

determine $m$ [130]—has been shown to be highly sensitive to small perturbations in pixel spectra and hyperparameter inputs [128]. In contrast, HySime is hyperparameter-free and can compute a high-quality, numerically stable estimate using only $X$ in just $O(D^2 n)$ operations.

### 2.3.2. Background on the AVMAX Algorithm

Alternating Volume Maximization (AVMAX) is a spectral unmixing algorithm, requiring $m$ as a parameter, that searches for vectors $\{u_i\}_{i=1}^m \subset \mathbb{R}^D$ that produce an $m$-simplex of maximal volume, subject to the constraint that each $u_i$ lies in the convex hull of the dataset after Principal Component Analysis (PCA) dimensionality reduction: projecting pixel spectra onto the span of the first $m-1$ principal components [70]. This dimensionality reduction step is motivated by the fact that any vector in the affine hull of the $m$ endmembers can always be expressed as $y = \mathbf{C}\alpha + \frac{1}{n}\sum_{x \in X} x$, where $\mathbf{C} \in \mathbb{R}^{D \times (m-1)}$ is related to the first $m-1$ principal components of $X$ [70,131]; see Lemma 1 in [70] for details. Endmembers are optimized through multiple partial maximization procedures (i.e., keeping $m-1$ endmembers constant and optimizing for volume while varying the $m^{\text{th}}$ endmember) until convergence [70]. AVMAX has become popular for spectral unmixing because of its strong performance guarantees and the rigor behind its optimization framework [12,70]. Indeed, in a noiseless, linearly mixed dataset containing the optimal endmember set, if each partial maximization problem in AVMAX converges to a unique solution, AVMAX is guaranteed to converge to the optimal endmember set [70]. Moreover, AVMAX can easily be modified to make it robust to random initialization; one can run multiple replicates of AVMAX in parallel and choose the endmember set with the largest volume. Once endmembers are learned, abundances may be computed using a nonnegative least squares solver: $(\mathbf{A}_{i1}\ \mathbf{A}_{i2}\ \ldots\ \mathbf{A}_{im}) = \operatorname{argmin}_{a \in [0,\infty)^m} \|x_i - \sum_{j=1}^m a_j u_j\|_2^2$ for each $x_i \in X$ [132].

### 3. Diffusion and Volume Maximization-Based Image Clustering

In spectrally mixed HSIs, any one pixel may correspond to a spatial region that contains many materials [12,14]. Thus, even state-of-the-art algorithms for unsupervised material clustering may perform poorly on mixed HSIs, failing to recover clusterings that can be linked to materials within the scene. Algorithms that do not directly incorporate a spectral unmixing step into their labeling may assign clusters corresponding to groups of materials rather than clusters corresponding to individual materials. Thus, additional improvements are needed to develop algorithms suitable for material clustering on mixed HSIs.

This section introduces the Diffusion and Volume maximization-based Image Clustering (D-VIC) algorithm (Algorithm 1) for unsupervised material clustering of HSIs. To learn material abundances, D-VIC first performs a spectral unmixing step: decomposing the HSI by learning the number of endmembers $m$ using HySime [128], implementing AVMAX with that $m$-value to learn endmembers [70], and calculating abundances and purity through a nonnegative least squares solver [132]. As will become clear in Section 4, this estimate for pixel purity resulted in high-quality material clustering with D-VIC. Nevertheless, the choice of the algorithm used for spectral unmixing in D-VIC is quite modular, and future work may consider applying other endmember extraction algorithms [12,92–94,130] and/or abundance solvers that explicitly constrain estimates to sum to one [133,134]. D-VIC then estimates empirical density using a kernel density estimate (KDE) defined by $p(x) = \frac{1}{Z}\sum_{y \in NN_N(x)} \exp(-\|x - y\|_2^2/\sigma_0^2)$, where $NN_N(x)$ is the set of $N$ $\ell^2$-nearest neighbors of $x$ in $X$, $\sigma_0 > 0$ is a KDE scale controlling the interaction radius between points, and $Z$ is a constant normalizing $p(x)$ so that $\sum_{y \in X} p(y) = 1$. By construction, $p(x)$ will be large if the pixel $x$ is close to its $N$ $\ell^2$-nearest neighbors in $X$ and small otherwise [18,42,135].

---

**Algorithm 1:** Diffusion and Volume maximization-based Image Clustering

---

**Input:** $X$ (HSI), $N$ (# nearest neighbors), $\sigma_0$ (KDE scale), $t$ (diffusion time), $K$ (# clusters)

**Output:** $\mathcal{C}$ (clustering)

1　Estimate $m$, the number of latent endmembers in $X$, using HySime [128];

2　Learn endmembers $\mathbf{U} \in \mathbb{R}^{m \times D}$ using AVMAX [70] and abundances $\mathbf{A} \in \mathbb{R}^{n \times m}$ using a nonnegative least squares solver [132];

3　For each $x \in X$, calculate pixel purity $\eta(x)$ and empirical density $p(x)$. Store $\zeta(x) = \frac{2\bar{p}(x)\bar{\eta}(x)}{\bar{p}(x)+\bar{\eta}(x)}$, where $\bar{p}(x) = \frac{p(x)}{\max_{y \in X} p(y)}$ and $\bar{\eta}(x) = \frac{\eta(x)}{\max_{y \in X} \eta(y)}$;

4　Build $d_t(x)$ using $\zeta(x)$, where diffusion distances are computed from a KNN graph with $N$ edges per pixel;

5　Assign $\mathcal{C}(x_{m_k}) = k$ for $1 \leq k \leq K$, where $\{x_{m_k}\}_{k=1}^K$ are the $K$ pixels maximizing $\mathcal{D}_t(x) = \zeta(x)d_t(x)$;

6　In order of non-increasing $\zeta(x)$, for each unlabeled $x \in X$, assign $x$ the label $\mathcal{C}(x^*)$, where $x^* = \text{argmin}_{y \in X}\{D_t(x,y)| \, \zeta(y) \geq \zeta(x) \text{ and } \mathcal{C}(y) > 0\}$.

---

To locate pixels that are both high-density and indicative of an underlying material, D-VIC calculates $\zeta(x) = \frac{2\bar{p}(x)\bar{\eta}(x)}{\bar{p}(x)+\bar{\eta}(x)}$, where $\bar{p}(x) = \frac{p(x)}{\max_{y \in X} p(y)}$ and $\bar{\eta}(x) = \frac{\eta(x)}{\max_{y \in X} \eta(y)}$. Thus, $\zeta(x)$ returns the harmonic mean of $p(x)$ and $\eta(x)$, which are normalized so that density and purity are approximately at the same scale. By construction, $\zeta(x) \approx 1$ only at high-density, highly pure pixels $x$. In contrast, if a pixel $x$ is either low-density or low-purity, then $\zeta(x)$ will be small. Importantly, $\zeta(x)$ downweights mixed pixels that, though high-density, correspond to a spatial region containing many materials. Thus, points with large $\zeta$-values will correspond to pixels that are modal (due to their high empirical density) and representative of just one material in the scene (due to their high pixel purity).

D-VIC uses the following function to incorporate diffusion geometry into its procedure for selecting cluster modes:

$$d_t(x) = \begin{cases} \max_{y \in X} D_t(x,y) & x = \text{argmax}_{y \in X} \zeta(y), \\ \min_{y \in X}\{D_t(x,y)|\zeta(y) \geq \zeta(x)\} & \text{otherwise.} \end{cases}$$

Thus, a pixel will have a large $d_t$-value if it is far in diffusion distance at time $t$ from its $D_t$-nearest neighbor of higher density and pixel purity. D-VIC assigns modal labels to the $K$ points maximizing $\mathcal{D}_t(x) = d_t(x)\zeta(x)$, which are high-density, high-purity pixels far in diffusion distance at time $t$ from other high-density, high-purity pixels.

After labeling cluster modes, D-VIC labels non-modal points according to their $D_t$-nearest neighbor of higher $\zeta(x)$-value that is already labeled. Importantly, D-VIC downweights low-purity pixels through $\zeta(x)$ in its non-modal labeling. Thus, pixel purity is incorporated in all stages of the D-VIC algorithm through $\zeta(x)$. D-VIC is provided in Algorithm 1, and a schematic is provided in Figure 2.

### 3.1. Computational Complexity

The computational complexity of the HySime algorithm is $O(D^2 n)$ operations [128], whereas the computational complexity of spectral unmixing using AVMAX and a standard nonnegative least squares solver [132] is $O((D^2 + m^4 + m^2 I)n)$ operations, where $I$ is the number of AVMAX partial maximizations [70]. We assume that nearest neighbor searches are performed using cover trees: an indexing data structure that enables logarithmic nearest neighbor searches [136]. To see this, define the doubling dimension of $X$ by $d = \log_2(c)$, where $c > 0$ is the smallest value for which any ball $B_p(p,r) = \{q \in X \mid \|p - q\| \leq r\}$ can be covered by $c$ balls of radius $r/2$. If the spectral signatures of pixels in $X \subset \mathbb{R}^D$ have doubling dimension $d$, a search for the $N$ $\ell^2$-nearest neighbors of each HSI pixel using cover trees has computational complexity $O(NDC^d n \log(n))$, where $C$ is a constant independent

of $n$, $D$, $N$, and $d$. Thus, if **W** is constructed using cover trees [136] with $N$ nearest neighbors, and $O(1)$ eigenvectors of **P** are used to calculate diffusion distances, then the computational complexity of D-VIC is $O((D^2 + m^4 + m^2 I)n + NDC^d n \log(n))$ [42,136].

So long as the spatial dimensions of the scene captured by an HSI are not changed, we expect that $m$ (the expected number of materials within the scene) will be constant with respect to the number of samples $n$. Similarly, numerical simulations have shown that if $m$ remains constant as the number of samples increases, then $I$ tends to grow only slightly [70]. If $m = O(1)$ and $I = O(\log(n))$ with respect to $n$, then the complexity of D-VIC reduces to $O(NDC^d n \log(n))$ (i.e., quasilinear in the image size).

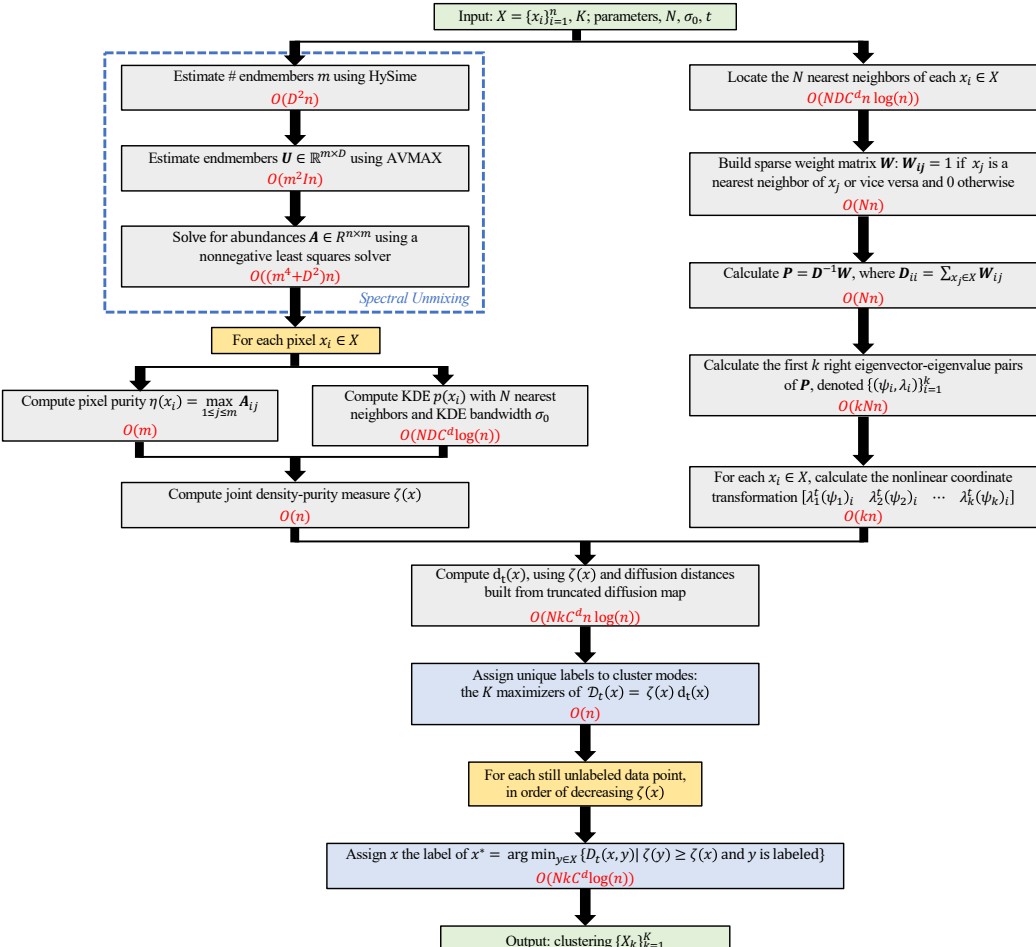

**Figure 2.** Diagram of the D-VIC clustering algorithm. The computational complexity of each step is colored in red. The scaling of D-VIC depends on $n$ (no. pixels), $D$ (no. spectral bands), $m$ (no. endmembers), $I$ (no. AVMAX maximizations), $N$ (no. nearest neighbors), $d$ (doubling dimension of $X$ [136]), and $C$: a constant independent of all other parameters [136]; see Section 3.1 for details. Note that all steps are quasilinear with respect to $n$, implying that D-VIC scales well to large HSI datasets. We remark that the spectral unmixing step (indicated with a blue box) is quite modular, and other approaches may be used in future work [12,90,93,94,130,133,134].

### 3.2. Comparison with Learning by Unsupervised Nonlinear Diffusion

An important point of comparison for D-VIC is the Learning by Unsupervised Nonlinear Diffusion (LUND) algorithm [18,42], which follows a similar procedure to D-VIC but crucially uses the KDE $p(x)$ in place of $\zeta(x)$. To give some motivation for why we advocate for $\zeta$ instead of $p$ for material clustering, we remark that for any one cluster, there may be multiple reasonable choices for cluster modes: pixels that are exemplary of the underlying cluster structure. In LUND, cluster modes are selected to be high-density pixels that are far in diffusion distance from other high-density pixels. However, not all high-density pixels

necessarily correspond to the underlying material structure. A maximizer of $p(x)$ could, for example, correspond to a spatial region containing a group of commonly co-occurring materials (rather than a single material). By weighting pixel purity and density equally, D-VIC avoids selecting such a pixel as cluster mode; thus, D-VIC modes will both indicate the underlying material structure and be modal, making these points better exemplars of underlying material structure than the modes selected by LUND.

We demonstrate this key difference between LUND and D-VIC by implementing both algorithms on a simple dataset (visualized in Figure 3) built to illustrate the idealized scenario where D-VIC outperforms LUND due to its incorporation of pixel purity. This dataset was generated by sampling $n = 5000$ points from an equilateral triangle in $\mathbb{R}^2$ centered at the origin with edge length $\sqrt{2}$; the $K = 3$ vertices of this triangle served as ground truth endmembers. We sampled 1000 data points from a Gaussian distribution with a standard deviation of 0.175 centered at each endmember, keeping only the samples lying within the convex hull of the ground truth endmember set. In addition, 2000 data points were sampled from a Gaussian distribution with zero mean and a smaller standard deviation of 0.0175. As such, high-purity points indicative of latent material structure were also relatively low-density, and density maximizers were engineered not to be indicative of latent material structure. Each point was assigned a ground truth label corresponding to its highest-abundance ground truth endmember.

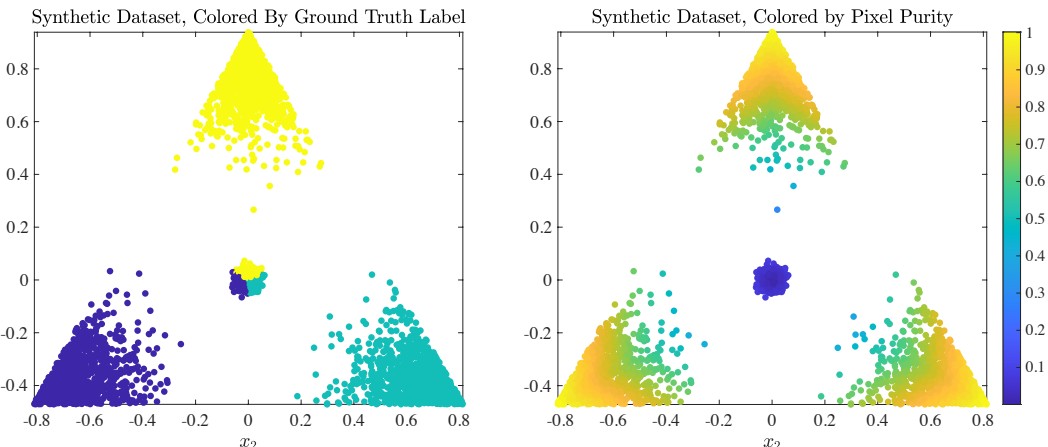

**Figure 3.** Ground truth labels and pixel purity of synthetic dataset sampled from a triangle in $\mathbb{R}^2$; the $K = 3$ vertices of this triangle served as ground truth endmembers. Notice that empirical density maximizers near the origin are also the lowest-purity data points.

For both LUND and D-VIC, overall accuracy (OA), defined to be the fraction of correctly labeled pixels, was optimized for across the same grid of relevant hyperparameter values (see Appendix A). The optimal clusterings and their corresponding OA values are provided in Figure 4. These results illustrate a fundamental limitation of relying solely on empirical density to select cluster modes in spectrally mixed HSI data. Because empirical density maximizers are not representative of the underlying material structure in this synthetic dataset, LUND cannot accurately cluster data points within the high-density, low-purity region near the origin, resulting in poor performance and an OA of 0.739. In contrast, D-VIC downweights high-density points that are not also high-purity and, therefore, selects points that are more representative of the dataset's underlying material structure as cluster modes. As a result, D-VIC correctly separates the high-density, low-purity region into three segments, yielding a substantially higher OA of 0.905, a difference of 0.166 when compared to LUND. We note that both LUND and D-VIC are related to classical spectral graph clustering methods [41,68,137] in their use of a diffusion process on the graph to learn the intrinsic geometry in the high-dimensional data, but differ in their use of data density (LUND) and data purity (D-VIC) in identifying cluster modes as well as in their use of an iterative labeling scheme.

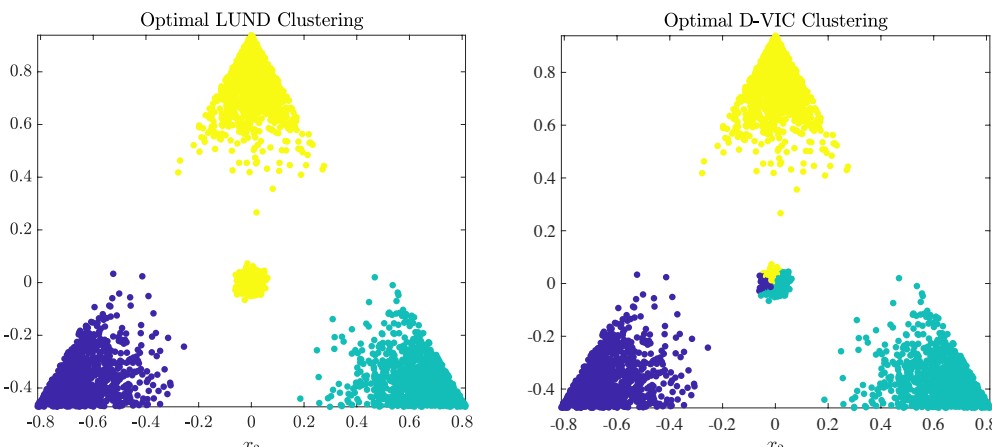

**Figure 4.** Optimal LUND (OA = 0.739) and D-VIC (OA = 0.905) clusterings of the synthetic dataset (Figure 3). D-VIC explicitly incorporates data purity into its labeling procedure, resulting in better clustering performance than LUND in the high-density, low-purity region near the origin.

## 4. Experiments and Discussion

This section contains a series of experiments indicating the efficacy of D-VIC. First, in Section 4.1, classical and state-of-the-art clustering algorithms were implemented on three real benchmark HSIs. D-VIC was compared against classical algorithms [17]: *K*-Means, *K*-Means applied to the first principal components of the HSI (*K*-Means+PCA), and GMM applied to the first principal components of the HSI (GMM+PCA). D-VIC was also compared against several state-of-the-art HSI clustering algorithms: Density Peak Clustering (DPC) [135], Spectral Clustering (SC) [41,137], Symmetric Nonnegative Matrix Factorization (SymNMF) [21], K-Nearest Neighbors Sparse Subspace Clustering (KNN-SSC) [19,20], Fast Self-Supervised Clustering (FSSC) [46] and LUND [18,42]. Our second set of experiments appears in Section 4.2, where D-VIC and other clustering algorithms were implemented on a remote sensing HSI generated over deciduous forest containing both healthy and dieback-infected ash trees in Madingley Village near Cambridge, United Kingdom [40,138].

In all experiments, the number of clusters was set equal to the ground truth $K$. Comparisons were made using OA and Cohen's $\kappa$ coefficient: $\kappa = \frac{p_o - p_e}{1 - p_e}$, where $p_o$ is the relative observed agreement between a clustering and the ground truth labels and $p_e$ is the probability that a clustering agrees with the ground truth labels by chance [139]. OA is a standard metric that, in some ways, captures the best sense of overall performance, as each pixel is considered of equal importance. However, it is biased in favor of correctly labeling large clusters at the expense of small clusters and can be misleading when a dataset has many small clusters of importance. To address this, we also consider $\kappa$; we note that in our experimental results, performance with respect to OA and $\kappa$ were highly correlated. OA was optimized for across hyperparameters ranging a grid of relevant values for each algorithm (see Appendix A). We report the median OA across 100 trials for *K*-Means, GMM, SymNMF, FSSC, and D-VIC to account for the stochasticity associated with random initial conditions. Diffusion distances were computed using only the first 10 eigenvectors of **P** in LUND and D-VIC. For D-VIC, AVMAX was run 100 times in parallel, and the endmember set that formed the largest-volume simplex was selected for later cluster analysis.

### 4.1. Analysis of Benchmark HSI Datasets

To illustrate the efficacy of D-VIC, we analyzed three publicly available, real HSIs often used as benchmarks for new HSI clustering algorithms; see Table 1 and Figure 5. Water absorption bands were discarded, and pixel reflectance spectra were standardized before analysis [140]. We clustered entire images but discarded unlabeled pixels when comparing clusterings to the ground truth labels. Below, each benchmark HSI analyzed in this section is overviewed in detail; see Table 1 for summary statistics on these benchmark HSIs.

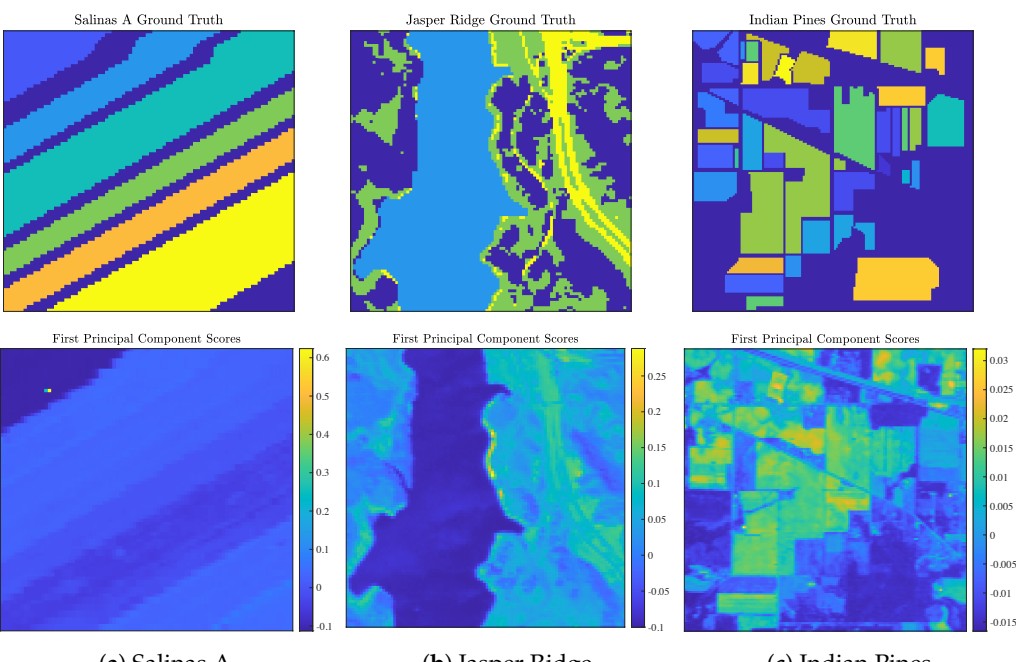

(**a**) Salinas A                          (**b**) Jasper Ridge                          (**c**) Indian Pines

**Figure 5.** Ground truth labels and first principal component scores for the real benchmark HSIs analyzed in this article: Salinas A (**a**), Jasper Ridge (**b**), and Indian Pines (**c**).

**Table 1.** Summary of benchmark HSI datasets analyzed in Section 4.1.

| Dataset | Spatial Resolution | Spectral Range | Spatial Dimensions | Num. Pixels | Num. Spectral Bands | Num. Clusters |
|---|---|---|---|---|---|---|
| Salinas A | 1.3 m | 380–2500 nm | $83 \times 86$ | $n = 7138$ | $D = 224$ | $K = 6$ |
| Jasper Ridge | 5.0 m | 380–2500 nm | $100 \times 100$ | $n = 10{,}000$ | $D = 224$ | $K = 4$ |
| Indian Pines | 20 m | 400–2500 nm | $145 \times 145$ | $n = 21{,}025$ | $D = 224$ | $K = 16$ |

1.  Salinas A (Figure 5a) was recorded by the Airborne Visible/Infrared Imaging Spectrometer (AVIRIS) sensor over farmland in Salinas Valley, California, USA, in 1998 at a spatial resolution of 1.3 m. Spectral signatures, ranging in recorded wavelength from 380 nm to 2500 nm across 224 spectral bands, were recorded across $83 \times 86$ pixels ($n = 7138$). Gaussian noise (with mean 0 and standard deviation $= 10^{-7}$) was added to each pixel to differentiate two pixels with identical spectral signatures. The Salinas A scene contains $K = 6$ ground truth classes corresponding to crop types.

2.  Jasper Ridge (Figure 5b) was recorded by the AVIRIS sensor over the Jasper Ridge Biological Preserve, California, USA, in 1989 at a spatial resolution of 5 m. Spectral signatures, ranging in recorded wavelength from 380 nm to 2500 nm across 224 spectral bands, were recorded across spatial dimensions of $100 \times 100$ pixels ($n = 10{,}000$). The Jasper Ridge scene contains $K = 4$ ground truth endmembers: road, soil, water, and trees. Ground truth labels were recovered by selecting the material of the highest ground truth abundance for each pixel.

3.  Indian Pines (Figure 5c) was recorded by the AVIRIS sensor over farmland in northwest Indiana, USA, in 1992 at a low spatial resolution of 20 m. Spectral signatures, ranging in recorded wavelength from 400 nm to 2500 nm across 224 spectral bands, were recorded across spatial dimensions of $145 \times 145$ pixels ($n = 21{,}025$). The Indian Pines scene contains $K = 16$ ground truth classes (e.g., crop types and manufactured structures) and many unlabeled pixels.

4.1.1. Discussion of Benchmark HSI Experiments

This section compares clusterings produced by D-VIC against those of related algorithms (Table 2). On each of the three benchmark HSIs analyzed, D-VIC produces a clustering closer to the ground truth labels than those produced by related algorithms. In the Indian Pines (Figure 5c) scene, pixels from the same class exist in multiple segments of the image, and the size of ground truth clusters varies substantially across the $K = 16$ classes. As such, though supervised and semi-supervised HSI classification algorithms may output highly-accurate classifications of the Indian Pines HSI [141–145], this image is expected to be challenging for fully-unsupervised clustering algorithms that rely on no ground truth labels. Nevertheless, D-VIC achieves higher performance than all other algorithms on this challenging dataset. Notably, though all other algorithms (including state-of-the-art algorithms, such as LUND) achieve $\kappa$-statistics in the same narrow range of 0.271 to 0.316, D-VIC achieves a substantially higher $\kappa$-statistic of 0.350. As such, incorporating pixel purity in D-VIC enables superior detection even in this difficult setting.

As visualized in Figure 6, D-VIC achieved nearly perfect recovery of the ground truth labels for Salinas A. Most notably, though all comparison methods erroneously separate the ground truth cluster indicated in yellow in Figure 5a (corresponding to 8-week maturity romaine), D-VIC correctly groups the pixels in this cluster, resulting in performance that was 0.089 higher in OA and 0.110 in $\kappa$ than the that of LUND, its closest competitor in Table 2. As such, downweighting high-density points that are not also exemplary of the latent material structure improves not only modal but also non-modal labeling. Moreover, what error does exist in the D-VIC clustering of Salinas A could likely be remedied through spatial regularization or smoothing post-processing [146,147].

**Table 2.** Performances of D-VIC and related algorithms on benchmark HSIs. The highest and second-highest performances are bolded and underlined, respectively. D-VIC offers substantially higher performance on all datasets evaluated.

|  | Salinas A | | Jasper Ridge | | Indian Pines | |
|---|---|---|---|---|---|---|
|  | OA | $\kappa$ | OA | $\kappa$ | OA | $\kappa$ |
| *K*-Means | 0.764 | 0.703 | 0.784 | 0.703 | 0.383 | 0.315 |
| *K*-Means + PCA | 0.764 | 0.703 | 0.785 | 0.703 | 0.382 | <u>0.316</u> |
| GMM + PCA | 0.611 | 0.512 | 0.789 | 0.701 | 0.364 | 0.292 |
| DPC | 0.629 | 0.529 | 0.809 | 0.727 | <u>0.410</u> | 0.271 |
| SC | 0.834 | 0.797 | 0.760 | 0.670 | 0.382 | 0.314 |
| SymNMF | 0.828 | 0.791 | 0.662 | 0.542 | 0.365 | 0.304 |
| KNN-SSC | 0.844 | 0.809 | 0.726 | 0.629 | 0.371 | 0.308 |
| FSSC | 0.830 | 0.793 | 0.780 | 0.691 | 0.396 | 0.281 |
| LUND | <u>0.887</u> | <u>0.860</u> | <u>0.815</u> | <u>0.737</u> | 0.404 | 0.312 |
| D-VIC | **0.976** | **0.970** | **0.865** | **0.805** | **0.445** | **0.350** |

D-VIC similarly achieved much higher performance than related state-of-the-art graph-based clustering algorithms on Jasper Ridge (as visualized in Figure 7). This difference in performance was substantially driven by the superior separation of the classes indicated in dark blue (corresponding to tree cover) and green (corresponding to soil) in Figure 5b. Indeed, though LUND groups most tree cover pixels with soil pixels in Figure 7, D-VIC correctly separates much of the latent structure for this class. The difference between LUND's and D-VIC's clusterings indicates that the pixels corresponding to the tree cover class, though lower density than pixels corresponding to the soil class have relatively high pixel purity.

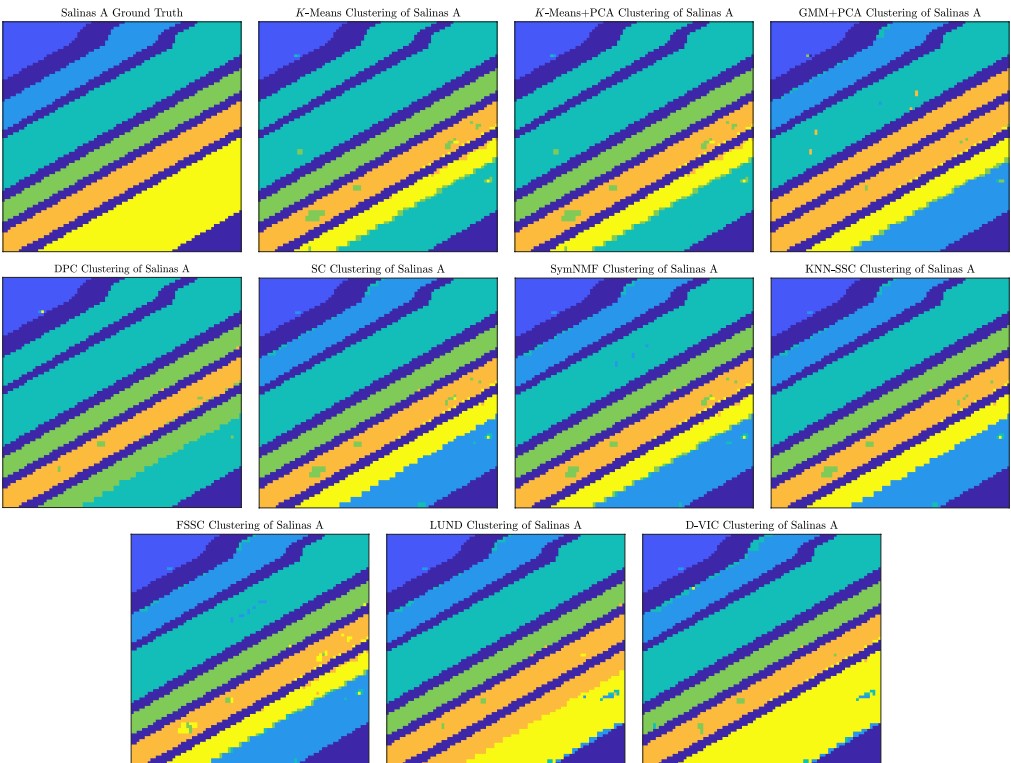

**Figure 6.** Comparison of clusterings produced by D-VIC and related algorithms on the Salinas A HSI. Unlike any comparison method, D-VIC correctly groups pixels corresponding to 8-week maturity romaine (indicated in yellow), resulting in the near-perfect recovery of the ground truth labels.

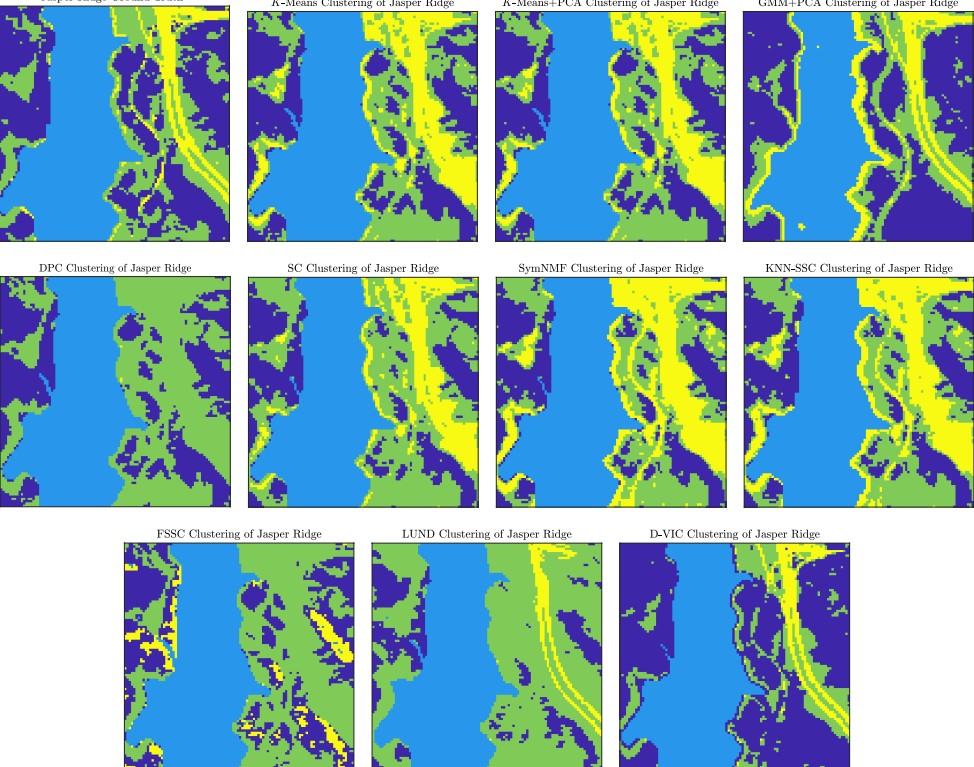

**Figure 7.** Comparison of clusterings produced by D-VIC and related algorithms on the Jasper Ridge HSI. D-VIC outperforms all other algorithms, largely due to superior performance among pixels corresponding to tree and soil classes (indicated in dark blue and green, respectively).

### 4.1.2. Runtime Analysis

This section compares runtimes of the algorithms implemented in Section 4.1.1, where hyperparameters were set to be those which produced the results in Table 2. All experiments were run in MATLAB on the same environment: a macOS Big Sur system with an 8-core Apple® M1™ Processor and 8 GB of RAM. Each core had a processor base frequency of 3.20 GHz. Runtimes are provided in Table 3. All classical algorithms have smaller runtimes than D-VIC, but the performances reported in Table 2 for these algorithms are substantially less than those reported for D-VIC. On the other hand, though KNN-SSC and SymNMF achieve performances competitive to D-VIC, unlike D-VIC, these algorithms appear to scale poorly to large datasets. DPC, which relies on Euclidean distances between high-dimensional pixel spectra, has lower runtimes on Salinas A than D-VIC but scales poorly to the larger Indian Pines image. In addition, D-VIC outperforms FSSC and operates at lower runtimes across HSI datasets. Finally, D-VIC outperforms LUND at the cost of only a small increase in runtime (associated with the spectral unmixing step).

**Table 3.** Runtimes (seconds) of D-VIC and related algorithms. D-VIC achieves runtimes comparable to state-of-the-art algorithms and scales well to the larger Indian Pines dataset.

|  | **Salinas A** | **Jasper Ridge** | **Indian Pines** |
|---|---|---|---|
| *K*-Means | 0.04 | 0.10 | 1.04 |
| *K*-Means + PCA | 0.10 | 0.14 | 0.58 |
| GMM + PCA | 0.13 | 0.23 | 2.19 |
| DPC | 3.20 | 6.41 | 25.77 |
| SC | 1.82 | 3.15 | 14.54 |
| SymNMF | 3.50 | 4.42 | 48.29 |
| KNN-SSC | 4.11 | 7.91 | 103.05 |
| FSSC | 13.53 | 30.40 | 130.72 |
| LUND | 2.35 | 4.14 | 14.74 |
| D-VIC | 4.95 | 7.64 | 23.70 |

### 4.1.3. Robustness to Hyperparameter Selection

This section analyzes the robustness of D-VIC's performance to hyperparameter selection. For each node in a grid of $(N, \sigma_0)$, D-VIC was implemented 50 times, and the median OA value across these 50 trials was stored. Performance degraded as $N$ increased substantially past 100, and such a choice is not advised. In Figure 8, we visualize how the performance of D-VIC varies with $N$ and $\sigma_0$. The relatively small range in nominal values of $\sigma_0$ in our grid reflects that pixels from the HSIs analyzed in this article are relatively close to their $\ell^2$-nearest neighbors on average. As is described in Appendix A—where our hyperparameter optimization is discussed in greater detail—the range of $\sigma_0$ used for each grid search is data-dependent, ranging the distribution of $\ell^2$-distances between pixel spectra and their 1000 $\ell^2$-nearest neighbors.

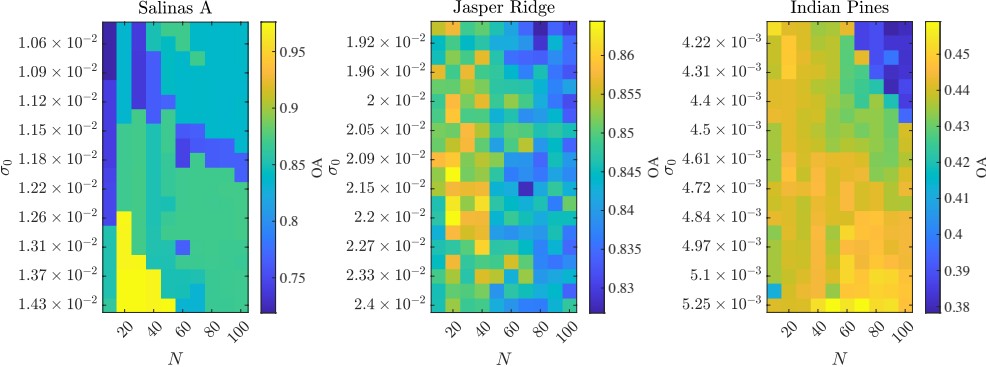

**Figure 8.** Visualization of D-VIC's median OA across 50 trials as hyperparameters $N$ and $\sigma_0$ are varied. D-VIC achieves high performance across a large set of hyperparameters.

It is clear from Figure 8 that D-VIC can achieve high performance across a broad range of hyperparameters on each HSI. Thus, given little hyperparameter tuning, D-VIC is likely to output a partition that is competitive with clusterings reported in Table 2. Figure 8 also motivates recommendations for hyperparameter selection to optimize the OA of D-VIC. Larger datasets (e.g., Indian Pines) tend to require larger values of $N$ for D-VIC to achieve high OA, corresponding with recommendations in the literature that $N$ should grow logarithmically with $n$ [42]. Additionally, D-VIC achieves the highest OA for datasets with high-purity material classes (e.g., Salinas A) using large $\sigma_0$. This reflects that, as $\sigma_0$ increases, the KDE $p(x)$ becomes more constant across the HSI and $\zeta(x) \approx \eta(x)$. Since purity is an excellent indicator of material class structure for Salinas A, D-VIC becomes better able to recover the latent material structure with larger $\sigma_0$.

We also analyze the robustness of D-VIC's performance to selection of the diffusion time parameter $t$. Using the optimal values of $N$ and $\sigma_0$, D-VIC was evaluated 100 times at $t$-values ranging $\{10, 20, \ldots, 200\}$. Figure 9, which visualizes the results of this analysis, indicates D-VIC achieves high OA values across a broad range of $t$; for each $t \in [90, 200]$, D-VIC outputs a clustering with OA equal to or very close to those reported in Table 2. These results indicate that D-VIC is well-equipped to provide high-quality clusterings given little or no tuning of $t$. Indeed, a simple choice of $t = 100$ works exceptionally well across all datasets.

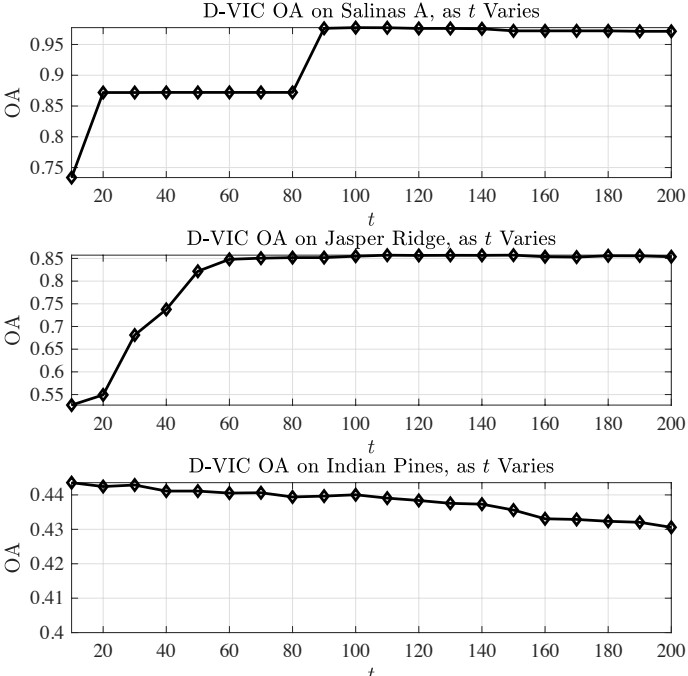

**Figure 9.** Analysis of D-VIC's performance as $t$ varies across $[10, 200]$. Values are the median OA across 100 implementations of D-VIC with the optimal $N$ and $\sigma_0$ values. Generally, $t$ appears to have little impact on the OA of D-VIC, and D-VIC achieves performances comparable to those reported in Table 2 across $t \in [90, 200]$, uniformly across all data sets.

### 4.2. Analysis of the Madingley HSI Dataset

This section presents implementations of D-VIC and other clustering algorithms on real HSI data to illustrate that unsupervised clustering algorithms may be used to generate ash dieback disease mappings from remotely-sensed HSI data, even when no ground truth labels are available. Algorithms were evaluated on the Madingley HSI dataset, which was collected by a manned aircraft in August 2018 over a 512 m × 356 m region of temperate deciduous forest in Madingley Village near Cambridge, United Kingdom [40]. This HSI was recorded by a Norsk Elektro Optikk hyperspectral camera (Hyspex VNIR 1800) at a high spatial resolution of 0.32 m. Spectral signatures, ranging in recorded wavelength

from 410 nm to 1001 nm across 186 spectral bands, were recorded across $1601 \times 1113$ pixels ($n = 1{,}816{,}835$). The Madingley HSI was preprocessed using QUick Atmospheric Correction [1] (to remove atmospheric effects on pixel spectra) and standardization of spectral signatures (to mitigate differences in illumination across pixels) [40].

Healthy and dieback-infected ash trees were identified in the Madingley scene using a pair of supervised classifiers [40]. First, to isolate ash tree crowns in the scene, a supervised Partial Least Squares Discriminant Analysis (PLSDA) classifier was trained to predict tree species using manually-collected ground truth labels for 166 tree crowns in the Madingley scene and 256 tree crowns in three other forest regions near Cambridge [40]. Labeled tree crowns were split into training (70%) and validation (30%) sets. The trained PLSDA classifier generalized well to the validation set, achieving an OA of 85.3% on those data [40]. Next, a supervised ash dieback disease map was generated for trees in the Madingley scene classified as ash by the PLSDA [40]. Specifically, a supervised random forest (RF) classifier was trained to classify a tree crown as one of three disease classes—healthy, infected, and severely infected—using the average pixel spectra from pixels corresponding to that tree crown. The RF was trained using manually-labeled tree crowns across the four aforementioned scenes and evaluated on a validation set consisting of 16 tree crowns from each disease class. The trained RF classifier was highly successful at identifying ash dieback disease, with an OA of 77.1% on its validation set [40]. Visualizations of the Madingley HSI and the RF disease mapping are provided in Figure 10.

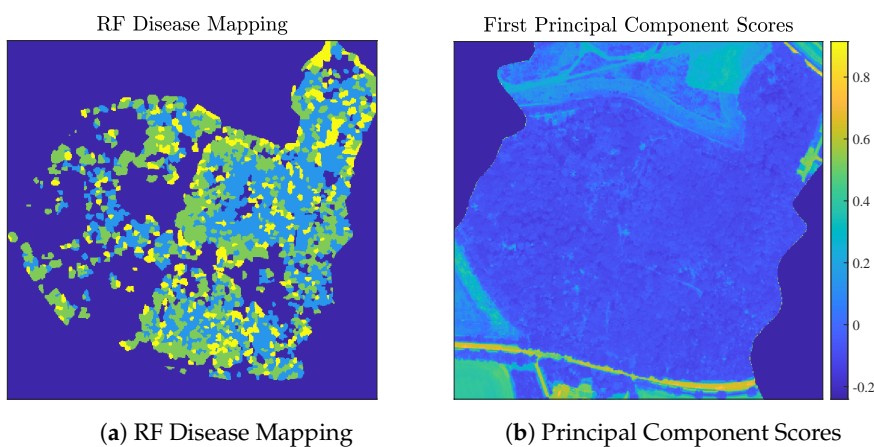

(**a**) RF Disease Mapping (**b**) Principal Component Scores

**Figure 10.** Visualizations of the Madingley HSI. The RF disease mapping is visualized in (**a**) and the Madingley HSI's first principal component scores are visualized in (**b**). In (**a**), yellow indicates severely-infected ash, green indicates infected ash, and light blue indicates healthy ash.

Dieback-infected ash trees tend to have a mosaic of healthy and dead branches, so bicubic interpolation [148] was implemented on the Madingley HSI before cluster analysis to downsample pixels to a 1.28 m spatial resolution [148]. Thus, each pixel covered a spatial region containing multiple branches, leading to a more holistic characterization of tree health (rather than of individual branches) [138]. Unsupervised clustering algorithms were evaluated on the $n = 72{,}775$ pixels in the resulting $401 \times 279$ scene corresponding to ash trees in the down-sampled PLSDA species mapping [40]. For each clustering algorithm, we set $K = 2$ so that clusters of pixels corresponded to healthy and dieback-infected trees. Unsupervised clusterings were evaluated by comparing against the supervised RF disease mapping after combining the "infected" and "severely infected" classes and aligning labels using the Hungarian algorithm.

Discussion of Madingley Experiments

Table 4 summarizes the overlap of D-VIC's and other algorithms' clusterings of the Madingley HSI with the RF disease mapping. Notably, four algorithms—SC, KNN-SSC, LUND, and D-VIC—achieved comparably high OA and $\kappa$ values. We remark that the RF disease mapping used for validation results from a supervised learning algorithm trained on a

small set of labels. Because it is an imperfect labeling of the Madingley HSI, small differences in OA or $\kappa$ values between SC, KNN-SSC, LUND, and D-VIC should not be taken as an indication that one of these clustering methods is better or worse than another. Nevertheless, these algorithms' high levels of overlap with the RF disease mapping indicate that graph-based unsupervised clustering algorithms like D-VIC may be applied to remotely-sensed HSI data to assess forest health even when ground truth labels are unavailable.

**Table 4.** Performances of D-VIC and related algorithms on the Madingley HSI. The highest and second-highest performances are bolded and underlined, respectively. Many graph-based algorithms—SC, KNN-SSC, LUND, and D-VIC—achieved approximately the same high performance on the Madingley HSI, indicating that graph-based HSI clustering algorithms may be used for unsupervised ash dieback disease mapping, even when no ground truth labels exist. KM denotes *K*-Means.

|     | KM | KM + PCA | GMM + PCA | DPC | SC | SymNMF | KNN-SSC | FSSC | LUND | D-VIC |
| --- | --- | --- | --- | --- | --- | --- | --- | --- | --- | --- |
| OA | 0.570 | 0.570 | 0.477 | 0.555 | 0.595 | 0.630 | **0.651** | 0.608 | <u>0.648</u> | 0.645 |
| $\kappa$ | 0.245 | 0.245 | 0.099 | 0.000 | <u>0.300</u> | 0.243 | **0.328** | 0.262 | 0.296 | 0.287 |

Though many graph-based HSI clustering algorithms exhibited similar levels of overlap with the supervised RF disease mapping, substantial differences exist between the unsupervised disease mappings obtained by different clustering algorithms, as can be observed in Figure 11. Indeed, LUND and D-VIC tended to predict ash dieback disease in regions considered healthy according to the RF disease map [40]. On the other hand, other similarly-performing graph-based clustering algorithms (SC and KNN-SSC) tended to label trees as healthy even in regions where the RF disease map indicates substantial dieback disease infection [40]. All clusterings exhibit salt-and-pepper error, indicating that spatial regularization [146,147] or majority voting within tree crowns [40,138] may improve overlap between unsupervised clusterings and the RF labeling even further.

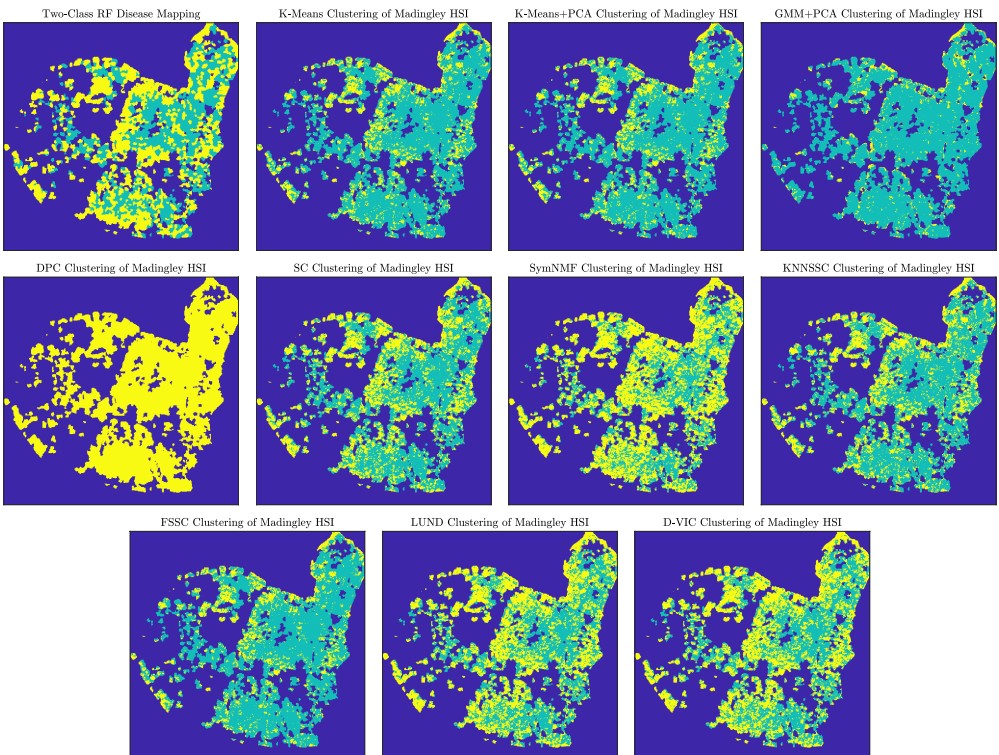

**Figure 11.** Comparison of clusterings produced by D-VIC and related algorithms on the Madingley HSI. Labels were aligned so that yellow indicates dieback-infected ash and teal indicates healthy ash. Though the performance of many graph-based algorithms (SC, KNN-SSC, LUND, and D-VIC) was similar in Table 4, qualitative differences exist between these algorithms' clusterings.

## 5. Conclusions

This article introduces the D-VIC clustering algorithm for unsupervised material classification in HSIs. D-VIC assigns modal labels to high-density, high-purity pixels within the HSI that are far in diffusion distance from other high-density, high-purity pixels [32,42,70]. We have argued that these cluster modes are highly indicative of underlying material structure, leading to more interpretable and accurate clusterings than those produced by related algorithms [18,42]. Indeed, experiments presented in Section 4 show that incorporating pixel purity into D-VIC results in clusterings closer to the ground truth labels on three benchmark real HSI datasets of varying sizes and complexities and enables high-fidelity unsupervised ash dieback disease detection on remotely-sensed HSI data. As such, D-VIC is equipped to perform efficient material clustering on broad ranges of spectrally mixed HSI datasets.

Future work includes modifying the spectral unmixing step in D-VIC. To demonstrate the effect of including a pixel purity estimate in a diffusion-based HSI clustering algorithm, we have chosen a simple, standard linear unmixing procedure to generate the pixel purity estimate in D-VIC: using HySime to estimate the number of endmembers in an HSI [128], AVMAX to estimate those endmembers [70], and a nonlinear least squares solver [132] to calculate abundances. The spectral unmixing procedure in D-VIC is quite modular, however, and improvements to D-VIC's clustering performance may be gained through improvements to this procedure; for example, by explicitly constraining abundances to sum to one [134] or accounting for nonlinear mixing of endmembers [123–125]. Linear endmember extraction is computationally inexpensive [70,128,132], and it results in a strong performance in D-VIC, but recent years have brought significant advances in algorithms for the nonlinear spectral unmixing of HSIs [123–125]. Modifying the spectral unmixing step in D-VIC may improve performance, especially for HSIs in which assumptions on linear mixing do not hold [123–125].

Additionally, much of the error in D-VIC's clusterings may be corrected by incorporating spatial information into its labeling. Such a modification of D-VIC may improve performance on datasets with spatially homogeneous clusters [60,146,149–154]. Moreover, it is likely that varying the diffusion time parameter $t$ in D-VIC may enable the detection of multiple scales of latent cluster structure, a problem we would like to consider further in future work [28,52,60,155]. Additionally, we expect D-VIC may be modified for active learning, wherein ground truth labels for a small number of carefully selected pixels are queried and propagated across the image [147,156]. Finally, we expect that D-VIC (or one of the extensions described above) may be modified for change detection in remotely-sensed scenes [86].

**Author Contributions:** Conceptualization, S.L.P. and K.C.; methodology, S.L.P., A.H.Y.C. and J.M.M.; software, S.L.P., A.H.Y.C. and K.C.; validation, S.L.P., K.C. and J.M.M.; formal analysis, S.L.P. and K.C.; investigation, S.L.P., K.C., A.H.Y.C., D.A.C. and J.M.M.; resources, J.M.M. and D.A.C.; data curation, J.M.M., A.H.Y.C. and D.A.C.; writing—original draft preparation, S.L.P.; writing—review and editing, K.C., A.H.Y.C., D.A.C., R.J.P. and J.M.M.; visualization, S.L.P.; supervision, R.J.P., D.A.C. and J.M.M.; project administration, R.J.P.; funding acquisition, J.M.M. All authors have read and agreed to the published version of the manuscript.

**Funding:** The US National Science Foundation partially supported this research through grants NSF-DMS 1912737, NSF-DMS 1924513, and NSF-CCF 1934553.

**Institutional Review Board Statement:** Not applicable.

**Informed Consent Statement:** Not applicable.

**Data Availability Statement:** Real benchmark hyperspectral image data used in this study can be found at the following links: http://www.ehu.eus/ccwintco/index.php?title=Hyperspectral_Remote_Sensing_Scenes (accessed on 12 December 2021) and https://rslab.ut.ac.ir/data (accessed on 12 December 2021). Experiments for benchmark data can be replicated at https://github.com/sampolk/D-VIC. Software and data required to replicate experiments on the Madingley HSI shall be made available upon reasonable request.

**Acknowledgments:** We thank C. Schönlieb and M. S. Kotzagiannidis for conversations that aided in the development of D-VIC. We acknowledge C. Barnes and 2Excel Geo for collecting the Madingley HSI used in this study. We thank the University of Cambridge for access to the Madingley field site and the Wildlife Trust for Bedfordshire, Cambridgeshire & Nottinghamshire for access to other forest field sites. Finally, we thank N. Gillis, D. Kuang, H. Park, C. Ding, M. Abdolali, D. Kun, I. Gerg, and J. Wang for making code for HySime [128], AVMAX [70,157], SymNMF [21], KNN-SSC [19], and FSSC [46] publicly available. We also thank the academic editor and the three reviewers for their helpful comments, which improved the presentation of this paper.

**Conflicts of Interest:** The authors declare no conflict of interest. The funders had no role in the design of the study; in the collection, analyses, or interpretation of data; in the writing of the manuscript, or in the decision to publish the results.

## Appendix A. Hyperparameter Optimization

This appendix describes the hyperparameter optimization performed to generate numerical results. The parameter grids used for each algorithm are summarized in Table A1. For *K*-Means + PCA and GMM + PCA, we clustered the first $z$ principal components of the HSI, where $z$ was chosen so that 99% of the variation in the HSI was maintained after PCA dimensionality reduction. Thus, *K*-Means, *K*-Means + PCA, and GMM + PCA required no hyperparameter inputs. For stochastic algorithms with hyperparameter inputs (SC, SymNMF, FSSC, and D-VIC), we optimized for the median OA across 10 trials at each node in the hyperparameter grids described below.

**Table A1.** Hyperparameter grids for algorithms. The number of nearest neighbors $N$ took values in $\mathcal{N}$: an exponential sampling of the set $[10, 900]$. The set $\mathscr{A}$ is a grid of values ranging from 0 to 1 used as FSSC regularization parameters. The set $\mathscr{D}$ contains $\ell^2$-distances between data points and their 1000 $\ell^2$-nearest neighbors. The set $\mathscr{T}$ is an exponential sampling of the diffusion process: $\mathscr{T} = \{0, 1, 2, \ldots, 2^2, \ldots, 2^T\}$. A "—" indicates a lack of a hyperparameter input.

|  | Parameter 1 Grid | Parameter 2 Grid | Parameter 3 Grid |
|---|---|---|---|
| *K*-Means | — | — | — |
| *K*-Means + PCA | — | — | — |
| GMM + PCA | — | — | — |
| DPC [135] | $N \in \mathcal{N}$ | $\sigma_0 \in \mathscr{D}$ | — |
| SC [41] | $N \in \mathcal{N}$ | — | — |
| SymNMF [21] | $N \in \mathcal{N}$ | — | — |
| KNN-SSC [19,20] | $N \in \mathcal{N}$ | $\lambda = 10$ | — |
| FSSC [46] | $N \in \mathcal{N}$ | $\alpha_u \in \mathscr{A}$ | $\ell = 2^{11}$ |
| LUND [42] | $N \in \mathcal{N}$ | $\sigma_0 \in \mathscr{D}$ | $t \in \mathscr{T}$ |
| D-VIC | $N \in \mathcal{N}$ | $\sigma_0 \in \mathscr{D}$ | $t \in \mathscr{T}$ |

All graph-based algorithms relied on adjacency matrices built from sparse KNN graphs. The number of nearest neighbors was optimized for each algorithm across $\mathcal{N}$: an exponential sampling of the set $[10, 900]$. KNN-SSC's regularization parameter was set to $\lambda = 10$, motivated by prior work with this parameter [19]. FSSC was evaluated using regularization parameters $\alpha_u \in \mathscr{A} = \{0, 10^{-5}, 10^{-3}, 10^{-1}, 0.5, 0.99, 0.999, 0.9999\}$, as was suggested in [46]. FSSC, as an anchor-based clustering algorithm, requires the number of anchor pixels $m$ as input. We set $\ell = 2^{11}$, as this $\ell$-value is greater than all $N \in \mathcal{N}$ [46]. We used the same KDE and hyperparameter ranges of $\sigma_0$ for DPC, LUND, and D-VIC. In our grid searches, $\sigma_0$ ranged $\mathscr{D}$: a sampling of the distribution of $\ell^2$-distances between data points and their 1000 $\ell^2$-nearest neighbors. Both LUND and D-VIC were implemented at time steps $t \in \mathscr{T} = \{0, 1, 2, 2^2, \ldots, 2^T\}$, where $T = \lceil \log_2 [\log_{\lambda_2(\mathbf{P})} (\frac{2 \times 10^{-5}}{\min(\pi)})] \rceil$. Searches end at time $t = 2^T$ because $\max_{x,y \in X} D_t(x, y) \leq 10^{-5}$ for $t \geq 2^T$ [52,60]. For each dataset, we chose the $t \in \{0, 1, 2, 2^2, \ldots, 2^T\}$ resulting in maximal OA. As is described in Section 4.1.3, D-VIC is quite robust to this choice of parameter.

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
