# Peer review of "Unsupervised Diffusion and Volume Maximization-Based Clustering of Hyperspectral Images"

_remotesensing, doi:10.3390/rs15041053_

Round 1

Reviewer 1 Report (New Reviewer)

In general, it is appropriate to support it with spelling and citations. However, it is not clear why Diffusion and Volume maximization-based Image Clustering (D-VIC) is needed in the study. Because, especially when looking at the results obtained in the Indian Pines data set, much less accuracy has been obtained when compared to the studies in the literature in which that data set is used. For this reason, it is important to compare the results with other studies in the literature for the widely used datasets (Salinas and Indian Pines) in the article. This comparison will be important in terms of revealing the necessity and functionality of the methods used in this study.

How the training and test parts of the datasets are separated should be explained clearly and precisely. At the same time, even if these rates are the same as the common usage in the literature, they will be more meaningful in terms of comparison.  

Author Response

See attached letter.

Reviewer 2 Report (Previous Reviewer 2)

The presented manuscript “remotesensing-2176113-peer-review-v1” is the modified previously presented after attending majority comments of all reviewers “remotesensing-2101962-peer-review-v2”. The text of this novel manuscript mostly repeats previous one using also much more references (147) against that used in previous manuscript. (119). This reviewer does not have comments for this manuscript.

Author Response

See attached letter.

Reviewer 3 Report (New Reviewer)

The manuscript sufficiently describes the methodology and results, so I think the paper would be almost ready for publication. However, I would like the authors my comments below for polishing it.

1. Specification of the dataset in Appendix B would be better to present as a summary table under Section 4 because readers may quickly refer details supporting the discussions.

Thank you in advance for your consideration.

Author Response

See attached letter.

Round 2

Reviewer 1 Report (New Reviewer)

Thanks to the authors.  Necessary corrections have been made.  The manuscript looks ready to be published.

This manuscript is a resubmission of an earlier submission. The following is a list of the peer review reports and author responses from that submission.

Round 1

Reviewer 1 Report

Please see the attached report.

Reviewer 2 Report

Due to an inherent trade-off between spectral and spatial resolution, many hyperspectral images are generated at a coarse spatial scale, and single pixels may correspond to spatial regions containing multiple materials. The authors introduced the Diffusion and Volume maximization-based Image Clustering (D-VIC) algorithm for unsupervised material discrimination. Incorporating pixel purity into its labeling procedure, D-VIC upweights pixels that correspond to a spatial region containing just a single material, yielding more interpretable clustering results. Novel D-VIC algorithm appears to demonstrate in numerical experiments better performance against state-of-the-art methods on a range of hyperspectral images, presenting well material discrimination and clustering of these data.

Comments:

 1) In opinion of this reviewer, the authors should implicitly present principal contributions of their novel Diffusion and Volume maximization-based Image Clustering (D-VIC) algorithm explaining their difference and performance in comparison with existing ones.

  2)  Another drawback of this manuscript, in opinion of this reviewer, which presents difficulties for understanding by potential reader, is the chosen structure of this manuscript via presenting different parts of the paper in numerous appendixes. This reviewer thinks that authors should select some important parts from the appendixes (Appendixes A, D, E) and present them into the main text of the manuscript. Other parts of these appendixes (for example, Appendix B. Details on Related HSI Clustering Algorithms) have common theoretical knowledge, and they are not important for understanding principal ideas of this manuscript and could be drop.

Reviewer 3 Report

The authors present a new clustering method they call “Diffusion and Volume maximization-based Image Clustering” (D-VIC), specifically for unsupervised segmentation of hyperspectral imagery. They combine multiple existing techniques like HySime and AVMAX as well as other concepts like KDE and diffusion geometry to obtain a stronger, more robust method.

The manuscript is well-written, well-structured and quite comprehensive. The authors explain the reasoning and background well and describe the thought process behind the algorithm in some detail. They also provide a comprehensive comparison with other classical and state-of-the-art clustering techniques based on multiple datasets.
Below, I list my main comments. Line numbers are indicated at the beginning of each comment.

1. The introduction section is very good, however, I would recommend shortening it. Especially in combination with section 2 (background) there are about 6 pages of different kinds of introductory topics. Although it is good to provide the reader with a solid overview, the authors should consider shortening both section 1 and 2. Instead, some parts of the annex may be interesting to include in the main text and further emphasis should be put on section 3 (see next point).
2. Section 3: I understand that you cover most of the details in section 2 but since this is supposed to be the main methodology section, it might be considered a bit short and superficial. This section should contain the main reasoning behind using different parts of the algorithm and why you consider certain elements from other techniques.
3. Section 4: please consider renaming this section and possibly splitting it into two (results and discussion), if nexessary.
4. 335ff: you focus your entire analysis on only two metrics (OA and κ). OA is generally not a very reliable metric for classification quality. It would make sense to provide additional metrics or at least justify your decision to focus only on these two.
5. 382ff: you mention that all your code was implemented in Matlab. Are you planning to provide the algorithm in an open source implementation (e.g. in Python)? Or at least as a reusable Matlab tool?
6. 385: just out of curiosity, do you expect performances being affected by the ARM architecture of the M1 chipset? Do you think that this architecture might carry advantages/disadvantages for algorithm performance compared to the more common x86 chips?
7. Please revisit the annexes, esp. annex B and E, and consider moving some of the insights to the main text. For example, it might make sense to shorten section 2 (see point 1) and instead add some more details on the specific differences/advantages of your new method in comparison to other existing ones (section 3).

Round 2

Reviewer 2 Report

The authors have attended all comments of this reviewer